# Lateral Flank Approach for Ovariohysterectomy in a Lion (*Panthera leo*) with a Ruptured Pyometra

**DOI:** 10.3390/vetsci8110245

**Published:** 2021-10-20

**Authors:** Taesik Yun, Jeongho Kim, Hyun-Gu Kang

**Affiliations:** 1Laboratory of Veterinary Internal Medicine, College of Veterinary Medicine, Chungbuk National University, Cheongju 28644, Chungbuk, Korea; yts87@hanmail.net; 2Cheongju Zoo, Cheongju 28311, Chungbuk, Korea; in-africa@korea.kr; 3Laboratory of Veterinary Obstetrics, College of Veterinary Medicine, Chungbuk National University, Cheongju 28644, Chungbuk, Korea

**Keywords:** flank, laparotomy, lion, ovariohysterectomy, *Panthera leo*, pyometra

## Abstract

An 8.5-year-old intact female lion (*Panthera leo*) with a history of vomiting, lethargy, and anorexia was referred to our institution. On physical examination, the lion weighed 180 kg and had a rectal temperature of 40 °C. Blood analysis showed mild neutrophilic leukocytosis, and abdominal ultrasonography revealed an enlarged uterus with echogenic fluid. Based on the clinical signs and hematologic and ultrasonographic findings, the lion was tentatively diagnosed with pyometra. Ovariohysterectomy was performed using the lateral flank approach to avoid complications that can occur with ventral celiotomy. Surgery was performed successfully despite unexpected rupture of the uterus which had occurred before the surgery. The lion recovered uneventfully and continued to do well at the 1 year follow up after surgery. To the best of our knowledge, this is the first reported case of ovariohysterectomy with the lateral flank approach in a lion with pyometra. Our report suggests that ovariohysterectomy with the lateral flank approach could be considered as an alternative method for pyometra in wild big cats with risk of complications at the surgical site.

## 1. Introduction

Pyometra is an acute or chronic clinical disease characterized by purulent exudate caused by secondary bacterial infection in the uterine lumen [1]. The exact pathogenesis of pyometra is not fully understood, but it is associated with both bacterial infection and hormonal factors [2,3,4]. Early detection and treatment could increase the survival rate, because pyometra can lead to endotoxemia, sepsis, and subsequent death.

Ovariohysterectomy is the most effective treatment option because the entire source of the infection is removed. It is commonly performed through either a ventral midline or lateral flank approach [5,6,7,8]. Even though the ventral midline approach is preferable to the lateral flank approach in ovariohysterectomy, each method has its advantages and disadvantages [5,6,7,8]. Compared with the ventral midline approach, in the lateral flank approach, it is more difficult to manipulate the distended uterus in pyometra or pregnancy because of limited exposure of the abdomen [6]. Therefore, surgery with the lateral flank approach is considered contraindicated in pyometra [6]; however, it could be an alternative for better wound healing and decreased potential risk of rupture at the surgical site in wild big cats.

Pyometra is a common disorder in small animals, such as dogs and cats [9,10]. However, there are only six reports of pyometra in lions (*Panthera leo*); ovariohysterectomy using the ventral midline approach was performed in three cases, necropsy in one case, and the type of surgery was not mentioned for two cases [11,12,13,14,15,16]; surgery with the lateral flank approach has not been reported. In the present case, instead of the ventral midline approach, surgery with the lateral flank approach was performed to reduce the potential risk of various complications at the surgical site.

In the present case, we show that ovariohysterectomy with the lateral flank approach could be considered in wild big cats with a risk of evisceration at the surgical site.

## 2. Case Presentation

An 8.5-year-old intact female lion (*Panthera leo*) with a history of vomiting, lethargy, and anorexia was referred to our institution. On physical examination under anesthesia using medetomidine (0.034 mg/kg, IM; SedaSTART, Yuhan, Seoul, Korea) and ketamine (1.67 mg/kg, IM; YUHAN KETAMINE 50 Inj., Yuhan, Seoul, Korea), the lion weighed 180 kg and showed a rectal temperature of 40 °C. Blood analysis showed mild neutrophilic leukocytosis (27.45 × 10^9^/L; mean of captive lions, 12.2 × 10^9^/L). Abdominal ultrasonography revealed a distended uterus with echogenic fluid (Figure 1). Based on the clinical signs and results of the above examinations, the lion was tentatively diagnosed with pyometra.

After 3 days, anesthesia was induced with medetomidine (0.034 mg/kg, IM) and ketamine (1.67 mg/kg, IM) by darting using a blowgun. Anesthesia induction was deemed complete by examining the lion for lack of response to stimulation upon lifting the front limbs or head after the darting. After anesthesia induction, the lion was blindfolded and transported in a car to the surgical room for surgical treatment. The lion was restrained on the operating table in the left lateral recumbent position. General anesthesia was maintained by inhalation of 3.5–4.0% sevoflurane (Sevofran, Hana Pharm Co., Ltd., Seoul, Korea) in 100% oxygen. Ketoprofen (1.67 mg/kg, IV; Eagle Ketoprofen 10% Inj., Eagle Vet. Tech Co., Ltd., Seoul, Korea), tramadol (1 mg/kg, IV; Trodon, Aju Pharm Co., Ltd., Seoul, Korea), and enrofloxacin (1.67 mg/kg, IM; Baytril 50 inj., Bayer Korea, Seoul, Korea) were administered preoperatively.

A right lateral flank laparotomy was performed via an approximately 20 cm incision from 5 cm below the 3rd lumbar transverse process towards the knee joint to facilitate uterine accessibility (Figure 2). To briefly summarize the abdominal wall incision procedure, the skin and peritoneum were sharply incised using a surgical knife, and the external abdominal oblique, transverse abdominal, and internal abdominal oblique muscles were bluntly separated. After the abdominal wall was incised, a hand was inserted into the abdominal cavity through the surgical incision to find the uterus and ovary. The abdominal cavity was filled with pus that had leaked from the ruptured uterus (Figure 3). On palpation, the uterus had the consistency of a sausage. Even though attempts were made to expose the right uterine horn as much as possible through an incision, a part of the uterine horn could not be exposed, and only the exposed part of the uterine horn was removed (a partial ovariohysterectomy, Figure 4). Removal of the left ovary and uterine horn was also performed in the same manner as on the right side. The remaining parts of the left and right uterine horns were sutured using the Parker-Kerr technique to prevent the drainage of pus. Immediately after suturing, 20 mL of enrofloxacin (Baytril 50 inj., Bayer Korea, Seoul, Korea) was administered to the uterine cavity and myometrium. Subsequently, after fixing four corner retractors to remove the leaked pus, the abdominal cavity was washed three times with warm physiological saline containing enrofloxacin (dilution of 5 mL of enrofloxacin in 1000 mL of normal saline), followed by three washes with physiological saline. The abdominal wall was sutured with No. 2 absorbent suture (SURGIFIT, AILEE Co., Ltd., Busan, Korea); the peritoneum with simple interrupted sutures, each muscle layer with continuous sutures, and the skin with subcuticular sutures with 1-0 absorbent suture. A transdermal fentanyl patch (1.1 µg/kg/hr; Durogesic D-trans, Janssen Cilag GmbH, Neuss, Germany), as a long-acting analgesic, was placed on the dorsal neck. After the completion of all procedures in the operating room, the lion was moved to the recovery room and the same volume of atipamezole (medetomidine antagonist; SedaSTOP, Yuhan, Seoul, Korea) was administered intramuscularly.

Cytologic analysis of a sample from the pus revealed bacterial infection with predominantly rod-shaped bacteria and numerous degenerative neutrophils (Figure 5). Two days after surgery, the lion recovered its appetite and activity. After 16 days, the surgical site had healed well (Figure 6). One year postoperatively, the lion was well, without clinical signs related to pyometra or septic peritonitis.

## 3. Discussion

Ovariohysterectomy for pyometra in lions is generally performed by ventral midline celiotomy. However, the ovariohysterectomy with the lateral flank approach could be used for the treatment of pyometra in a lion (*Panthera leo*) to avoid complications that may be caused by ventral midline celiotomy.

Pyometra is a common disease in small animals, but there are only limited reports in lions [11,12,13,14,15,16]. However, pyometra may be more common in wild animals, especially lions, than previously reported [13]. A study of pyometra in captive large fields showed that the incidence of pyometra in lions was 29.2% (7/24) and lions had a higher potential for pyometra than other large cat species (Tiger [*Panthera tigris*], 2.7%, 2/73; Cougar [*Felis concolor*], 0%, 0/11) [13]. In particular, the incidence of pyometra in lions over 10 years of age was 62.5% (5/8) [13]. Therefore, pyometra should be considered in intact females with anorexia or lethargy, especially in lions older than 10 years of age.

The ventral midline approach is a more common method than the lateral flank approach for ovariohysterectomy. However, in a study of complications following ovariohysterectomy in shelter-housed cats, the risk of wound complications was higher with the ventral midline approach than with the lateral flank approach [17]. Therefore, the lateral flank approach could be a better choice than the conventionally preferred ventral midline approach. Indications for the lateral flank approach include situations where postoperative care may be limited or excessive mammary development, such as enlargement of mammary glands due to lactation and mammary gland hyperplasia [6]. The main advantages of the lateral flank approach are that the potential for evisceration by dehiscence is reduced and the surgical wound can be observed from a distance [6]. These advantages are particularly important when treating wild animals with limited postoperative examinations. Furthermore, when planning treatment for wild animals, safety issues should also be considered because of the potential for serious harm to the animal care staff. On the other hand, the lateral flank approach is generally contraindicated in several situations including obesity, uterine distention (pregnancy or pyometra), and young age (<12 weeks) [6]. In the present case, because of concerns about evisceration from the large amount of abdominal fat and organs and safety issues, despite the contraindication in pyometra, ovariohysterectomy with the lateral flank approach was performed successfully without any complications.

The transdermal fentanyl patch is primarily used for relief from postoperative pain and severe pain caused by tumors, aortic thromboembolism, pancreatitis, etc. [18]. They are commonly used in small animals; however, there has been no report of use of fentanyl patch in wild big cats. The recommended dosage varies, but the general range used in veterinary medicine is 1–5 µg/kg/h [18]. The dosage in the present case was approximately 1 µg/kg/h (200 µg/body/h), and it showed good pain control without any adverse effects.

In conclusion, ovariohysterectomy with the lateral flank approach was performed successfully despite a ruptured pyometra which had occurred before the surgery. The lion recovered uneventfully and continued to do well at more than one year after surgery. Even though this is a single case, our results suggest that ovariohysterectomy with the lateral flank approach could be considered as an alternative method for pyometra in wild big cats with risk of complications at the surgical site.

## Figures and Tables

**Figure 1 vetsci-08-00245-f001:**
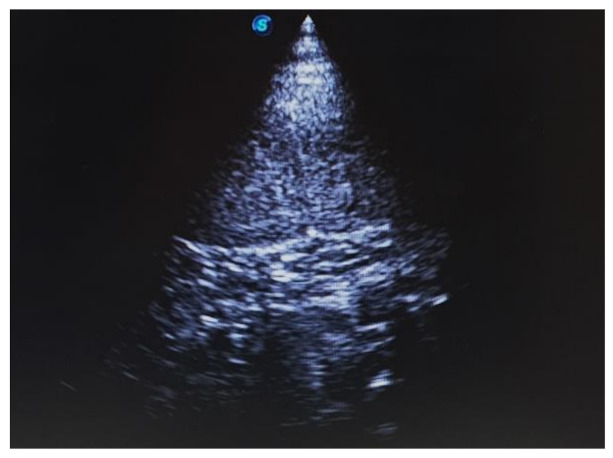
Uterine horn in a lion (*Panthera leo*) with pyometra. Presence of markedly distended uterine horn with echogenic intraluminal material.

**Figure 2 vetsci-08-00245-f002:**
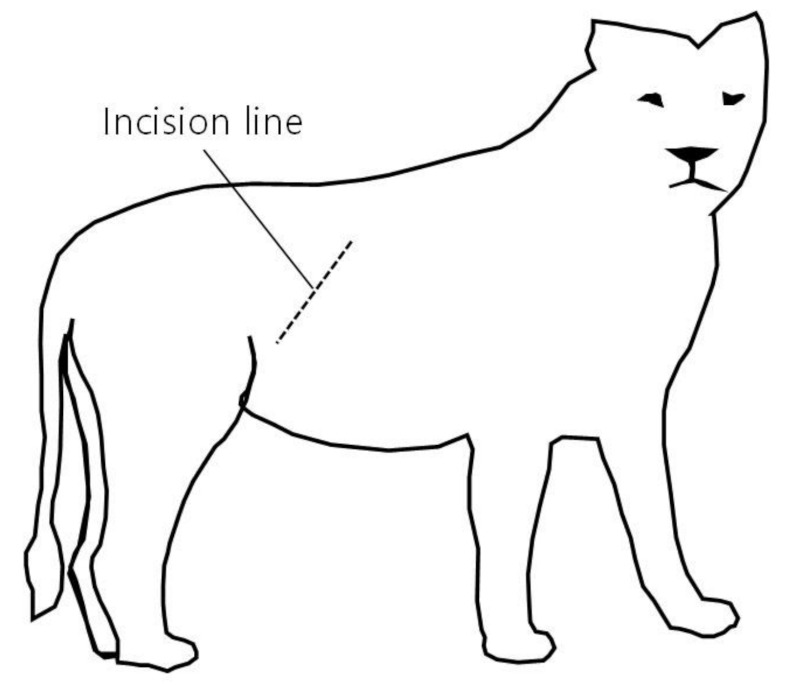
Skin incision line for pyometra surgery in a lion (*Panthera leo*). A right lateral flank laparotomy was performed via an approximately 20 cm incision from 5 cm below the 3rd lumbar transverse process towards the knee joint.

**Figure 3 vetsci-08-00245-f003:**
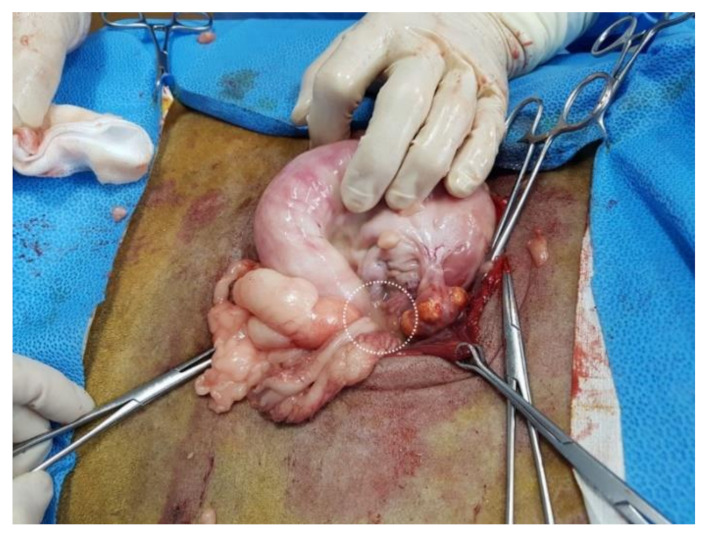
Intraoperative appearance of the uterus of a lion (*Panthera leo*) with pyometra. There is pus leakage (white dotted circle) around the uterus.

**Figure 4 vetsci-08-00245-f004:**
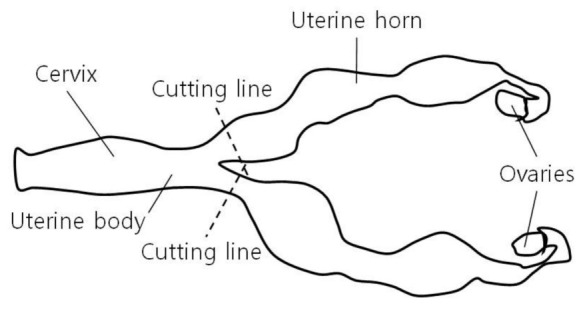
Cutting lines for removal of the uterine horns (spot lines). The uterine horns were removed as much as possible through the incision (a partial ovariohysterectomy). The parts of the left and right uterine horns remaining in the abdominal cavity were sutured using the Parker-Kerr technique.

**Figure 5 vetsci-08-00245-f005:**
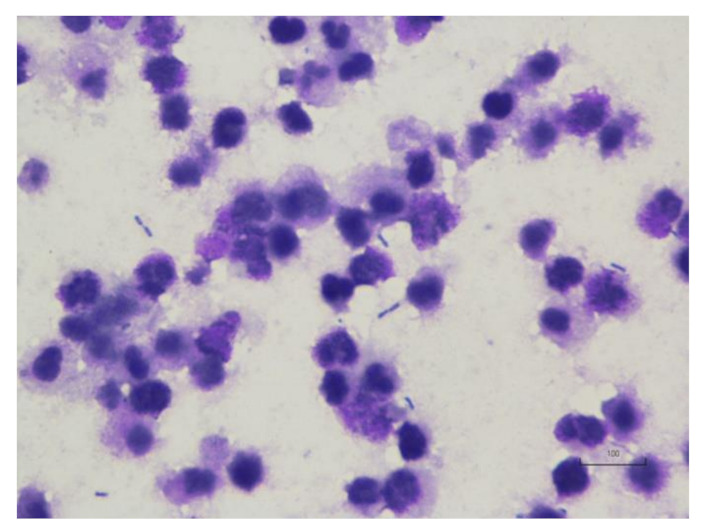
Cytology of pus of the uterus in a lion (*Panthera leo*) with pyometra. There are numerous degenerative neutrophils and many bacterial rods.

**Figure 6 vetsci-08-00245-f006:**
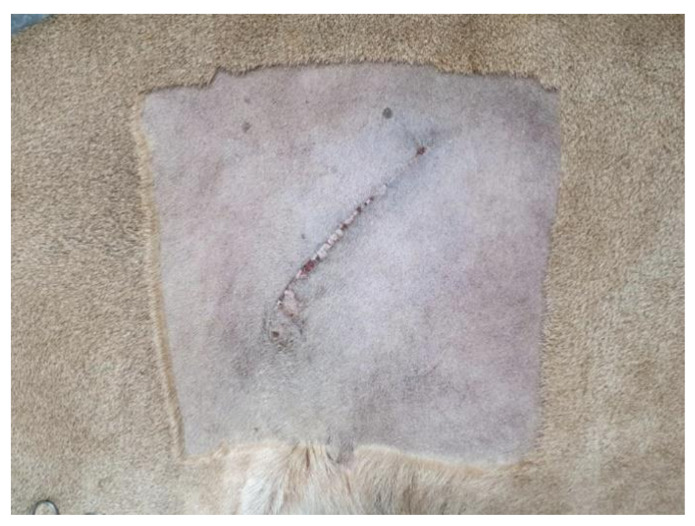
Surgical site 16 days after ovariohysterectomy with lateral flank approach.

## Data Availability

The data presented in this study are available in the manuscript.

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
