# Peer review of "Lateral Flank Approach for Ovariohysterectomy in a Lion (Panthera leo) with a Ruptured Pyometra"

_vetsci, 2021, doi:10.3390/vetsci8110245_

Round 1
Reviewer 1 Report
Below my comments to the manuscript ” Lateral flank approach for ovariohysterectomy in a lion (Panthera leo) with a ruptured pyometra”:
A case of ovariohysterctomy with lateral flank approach in a lion with ruptured pyometra is described. Surgery was performed succesfully and the lion recovered uneventfully. The Authors concluded that the lateral flank approach could be considered as an alternative metod for pyometra treatment in wild big cats.
Strengths
This is the first first report of a sucessful ovariohysterctomy with the lateral flank approach in a lion with pyometra.
Weknesses
Generally, for animals with pyometra the flank approach is not recommended because it does not provide sufficient exposure to manipulate the distended uterus.
The uterine rupture had not been diagnosed by previous ultrasound examination and it cannot be excluded that it occurred during the extraction of the uterus through a relatively short incision in the abdominal wall.
The technique of ovariohysterctomy is sparingly described.
Remarks:
What is meant by partial ovariohysterectomy? Please describe it in more detail and provide a drawing of the location of the ligatures and cuts.
Did the abdominal wall incision have to be extended?
Reviewer 2 Report
Excessive repetition of nearly identical sentences takes away from the article.
Lines 19 & 20: “To the best of our knowledge, this is the first reported case of ovariohysterectomy with the lateral flank approach in a lion with pyometra.”
Lines 47 & 48: “To the best of our knowledge, this is the first reported case of ovariohysterectomy with the lateral flank approach for pyometra in a lion…”
Lines 112 & 113: “To the best of our knowledge, this is the first case report describing the lateral flank approach for the treatment of pyometra in a lion.”
Lines 148 & 149: “To the best of our knowledge, this is the first reported case of ovariohysterectomy with the lateral flank approach in a lion with pyometra.”
The description of the surgery is incomplete. A detailed description of the surgical procedure would be helpful.
- “The abdominal wall was incised in the direction of the posterior lower abdomen…”This sentence is confusing. After reading the description several times, I still don’t know what this sentence means.
- Line 77: “Each muscle layer was bluntly separated.”What are the muscles layers? Were they bluntly separated as this sentence says or were they incised as the sentence above says?
- Line 81: “The right uterus was exposed as much as possible…”Suspecting that the lion does not have two uteruses I assume the authors mean the right uterine horn.
- Line 81: … “a partial ovariohysterectomy was performed.”How? Describe what was done>
- Line 82: “The left uterus was also subjected to…”Suspecting that the lion does not have two uteruses I assume the authors mean the left uterine horn.
- Line 82 & 83: “… was also subjected to a partial ovariohysterectomy using the same method.”What method? The method was never described.
- Line 83:“The remaining uterus was sutured the Parker -Kerr technique.” As best I can determine from the surgical description part of the right uterine horn, part of the left uterine horn and all of the uterine body were left in the abdomen of the lion. I don't believe this is appropriate when managing a case of pyometra?
Round 2
Reviewer 2 Report
Please clarify if the uterus was ruptured prior to surgery or during surgery. Lines 82 & 83, “The abdominal cavity was filled with pus that had leaked from the ruptured uterus,” makes it sound like the uterus was ruptured prior to surgery. But Lines 166 & 167, “In conclusion, ovariohysterectomy with the lateral flank approach was performed successfully despite unexpected rupture of the pyometra,” makes it sound like the uterus was ruptured during the surgery. If the uterus was ruptured during surgery the current title of the article, “Lateral Flank Approach for Ovariohysterectomy in a Lion (Panthera leo) with a Ruptured Pyometra,” is misleading. If the uterus ruptured during surgery a more appropriate title would be “Lateral Flank Approach for Ovariohysterectomy in a Lion (Panthera leo) with Pyometra,”
Line 34: Not sure “neutralization” is the best choice of words.
Line 37 &3 8: Suggest this wording: “Therefore, surgery with the lateral flank approach is considered contraindicated in pyometra:”
Line 67: Suggest replacing the word “arms” with “front limbs.”
Line 81: Suggest replacing the word “section” with “incision.” Making the sentence read “a hand was inserted into the abdominal cavity through the surgical incision…”
